# Carbon Nano-Fiber/PDMS Composite Used as Corrosion-Resistant Coating for Copper Anodes in Microbial Fuel Cells

**DOI:** 10.3390/nano11113144

**Published:** 2021-11-21

**Authors:** Fatma Bensalah, Julien Pézard, Naoufel Haddour, Mohsen Erouel, François Buret, Kamel Khirouni

**Affiliations:** 1Laboratoire Ampère, Ecole Centrale de Lyon, 36 Avenue Guy de Collongue, 69134 Ecully, France; fatma.bensalah@ec-lyon.fr (F.B.); julien.pezard@ec-lyon.fr (J.P.); francois.buret@ec-lyon.fr (F.B.); 2Laboratory of Physics of Materials and Nanomaterials Applied at Environment, Faculty of Sciences in Gabes, Gabes University, Gabes 6072, Tunisia; mohsen.erouel@gmail.com (M.E.); khirouni.kamel@gmail.com (K.K.)

**Keywords:** microbial fuel cell, anode materials, copper electrodes, CNF-PDMS, surface modification, biofilm

## Abstract

The development of high-performance anode materials is one of the greatest challenges for the practical implementation of Microbial Fuel Cell (MFC) technology. Copper (Cu) has a much higher electrical conductivity than carbon-based materials usually used as anodes in MFCs. However, it is an unsuitable anode material, in raw state, for MFC application due to its corrosion and its toxicity to microorganisms. In this paper, we report the development of a Cu anode material coated with a corrosion-resistant composite made of Polydimethylsiloxane (PDMS) doped with carbon nanofiber (CNF). The surface modification method was optimized for improving the interfacial electron transfer of Cu anodes for use in MFCs. Characterization of CNF-PDMS composites doped at different weight ratios demonstrated that the best electrical conductivity and electrochemical properties are obtained at 8% weight ratio of CNF/PDMS mixture. Electrochemical characterization showed that the corrosion rate of Cu electrode in acidified solution decreased from (17 ± 6) × 10^3^ μm y^−1^ to 93 ± 23 μm y^−1^ after CNF-PDMS coating. The performance of Cu anodes coated with different layer thicknesses of CNF-PDMS (250 µm, 500 µm, and 1000 µm), was evaluated in MFC. The highest power density of 70 ± 8 mW m^−2^ obtained with 500 µm CNF-PDMS was about 8-times higher and more stable than that obtained through galvanic corrosion of unmodified Cu. Consequently, the followed process improves the performance of Cu anode for MFC applications.

## 1. Introduction

Energy security and clean water have become major concerns in today’s world, requiring efficient technologies for renewable energy production and sustainable water treatment. Microbial fuel cell (MFC) is a promising renewable bioenergy technology using bacterial biofilms as biocatalysts to convert organic matter into electricity [1,2]. MFCs can operate with wastewater, industrial effluents, agricultural waste and sewage, to produce electrical energy. Numerous studies have been conducted in past decade for increasing power production and treatment efficiency of MFC technology and significant progress has been achieved [3,4,5,6,7]. However, the electricity production and removal efficiency of MFCs remain low for their implementation in real-world applications [7,8]. These performance limitations are mainly due to the high cost and low-quality of materials used as anodes in MFC. The anode is an important part of MFC because it is used as the final electron acceptor of the anaerobic respiratory chain for bacterial growth [9,10,11]. The development of low cost and high-performance anode materials are still critical challenges for the practical implementation of MFCs. MFC anode materials must have good biocompatibility, high resistance to corrosion, good mechanical stability and high electrical conductivity [12]. Carbon materials which are commonly used to make anodes of the MFC (graphite, graphene, carbon felt, have low electrical conductivity and high manufacturing costs [13]. Indeed, electrical conductivity of carbon materials (10^4^–10^5^ S m^−1^) is two to three orders of magnitude below that of pure metals [14]. The resistivity of carbon anodes causes power losses and drop in potential distribution on the electrode surface [15]. Thus, utilization of metallic materials as electrodes is the best way to improve electrical conductivity of anodes. Cu is an interesting choice as an anode material because it is one of the best metallic conductors of electricity (58 × 10^6^ S m^−1^) [14]. However, Zhu et al. demonstrated that Cu is an unsuitable anode material for MFC application [16]. The authors studied the performance of Cu as an anode and the maximum power density generated was similar to abiotic controls (without bacterial inoculation). The author concluded that Cu electrodes should not be used in MFCs as anodes. Indeed, Cu anodes suffer from poor biocompatibility and corrosion fatigue. The corrosion of Cu anodes affects negatively electricity production due to the toxicity of Cu ions to bacteria. Thus, surface modification is a necessary step when using Cu material as anode in MFC operations. To the best of our knowledge, only one recent study of Mwale et al., investigated the application of Cu coated with a polyaniline thin film as an anode material in MFC application [17]. In this study, polyaniline was used as a protector conducting polymer of Cu anodes. The highest power density performance obtained with polyaniline-Cu anodes was around 1.5 mW m^−3^. This study showed the possibility of producing electrical energy with Cu anode in MFC. However, the electrical performance obtained with this surface modification method is low. Furthermore, some factors currently limit the applications of electrochemically polymerized polymers such as polyaniline and polypyrrole because of their low mechanical properties and low processibility [18]. Until now, no studies have described other modification methods to modify the Cu surface and fabricate new cost-effective anodes in the MFCs to our knowledge.

In this study, Cu anode surfaces were modified with thin layer of low-cost and mechanical stable CNF-PDMS (Polydimethylsiloxane) coating (Figure 1). PDMS is an attractive polymer with physically and chemically stable properties [19]. Recently, we demonstrated that PDMS doped with CNF is a conducting polymer providing more significant electrical active sites for electron transfer between exoelectrogenic bacteria and stainless steel coated electrodes [19]. This coating method could be deposited on large electrode surfaces at low costs. Indeed, PDMS can be processed by the roll-to-roll method on flexible supports and produced in high numbers [20]. In this study, experiments were carried out to optimize properties of a flexible foil copper anode coated with CNF-PDMS polymer as a protective barrier. Structural, morphological, electrical and electrochemical properties of CNF-PDMS coated Cu anodes were studied for power generation in MFCs.

## 2. Materials and Methods

### 2.1. Preparation of CNF-PDMS Composite

CNF-PDMS composite was made using a 10:1 (*w*/*w*) mixture of PDMS base (Sylgard 182 Silicone Elastomer) and curing agent (Sylgard 182 Silicone Elastomer) degassed under vacuum. Carbon nanofibers (CNFs) with a diameter of 200 nm and length of 50 μm (purchased from Sigma-Aldrich Saint Quentin Fallavier, St. Quentin Fallavier Cedex, France), purity > 99.9% carbon basis) reinforced in PDMS polymer at different weight ratios of 2, 4, 6 and 8% (*w*:*w*) CNF/PDMS to prepare the conductive composite CNF-PDMS. The mixture is thoroughly mixed for 20 min by hand until obtaining a homogeneous paste.

### 2.2. Coating of Cu Electrodes with CNF-PDMS Composite

Copper foil (176–7501 purchased from Radiospares (Beauvais, France), France) was cut into electrodes with dimensions of 5 × 1 cm^2^. An overall area of 10.42 cm^2^ was determined for each electrode including both sides and foil thickness (35 μm). Cu electrodes were washed with acetone and ethanol before using and dried at room temperature. CNF-PDMS paste was then cast on Cu electrode surfaces, and the obtained layer was leveled to the desired height (250 μm, 500 μm, and 1000 μm) with patterns placed around the electrode. Cu coated electrodes were then cured at 80 °C for 2 h. The thickness of the formed CNF-PDMS layer on the flat surface of the electrodes was controlled using a mechanical profilometer (VeecoDektak 3030, (Veeco instruments, Dourdan, France)) and was around the pattern height.

### 2.3. Electrode Characterization

The morphology of the CNF-PDMS layers was examined with a scanning electron microscope (SEM, JEOL model JSM-7401F, (JOEL, Croissy, State abbreviation if possible (Such as USA and Canada), France)). The resistivity of this material was determined by four-point measurements with a probe station and a Keithley 4200 Source Measure Unit (SMU) (Tektronix, Les Ulis, France). Light microscopy was used to visualize the color and morphology of Cu electrodes after their utilization as anodes in MFC application. A microscope featuring long-working distance optics (Nikon Eclipse, LV 150, (Nikon France, Champigny Sur Marne, France)) was used. A DS-Fi2 Nikon digital camera (Nikon France, Champigny Sur Marne, France) was directly mounted on the microscope for acquiring images of the various surfaces. Cyclic voltamperometry (CV) was used for determination of the potential range, capacitive effect, active area and kinetics. The CV tests were carried out with potentiostat Origalys 0GF01A (Origalys, Rilleux-La-Pape, France). Coated and uncoated Cu electrodes were used as working electrodes vs. Ag/AgCl reference electrodes, and the current was collected by a Pt auxiliary electrode.

### 2.4. Corrosion Tests

The galvanic corrosion of coated and uncoated Cu electrodes was investigated by Tafelcurves plotted using linear sweep voltammetry (LSV). A three-electrode arrangement was utilized at a scanning rate of 10 mV s^−1^ in aqueous solutions acidified with sulfuric acid at pH of 3.00. A potentiostat OGS 500 (POrigalys, Rilleux-La-Pape, France) was used to perform electrochemical characterizations. Coated and uncoated Cu electrodes were used as working electrodes, a commercial saturated Ag/AgCl electrode as a reference and a Pt wire electrode as an auxiliary electrode. The geometric surface area of metal electrodes was used for calculating galvanic corrosion current density. 

### 2.5. MFCs Setup and Operation

Single-chamber batch MFCs were set up in 250 mL Wheaton bottles in the laboratory at ambient temperature (Appendix A). The anode consisted of one 10 × 15 cm piece of carbon cloth. The carbon cloth would be cut during sampling to have 25 (0.5 × 8 cm^2^) pieces. The cathode was prepared with PTFE coating and 5% of platinum catalyst as previously described [1]. The MFCs were filled with 250 mL of primary effluent and 1 g of dehydrated sludge from a Grand Lyon domestic wastewater treatment plant (Lyon, France) and fed with 1 g L^−1^ of sodium acetate. The experiment was completed in triplicate. MFCs were started with external resistances of 330 ohms.

### 2.6. Polarization Curve Measurement

The polarization curves of MFCs were measured by Linear Seep Voltammetry (LSV) utilizing a two-electrode arrangement with a potentiostat (OGS 500 from Origalys, (Origalys, Rilleux-La-Pape, France)). LSV was performed from the open circuit potential to 0 V using a scan rate of 10 mV s^-1^. The power was calculated by multiplying the current by the voltage. The geometric surface area of anodes was used for calculating power density of MFCs. 

## 3. Results and Discussion

### 3.1. Characterization of CNF-PDMS-Copper Electrodes for Different Doping Levels

To assess the correlation between the composition of the coating layers and their electrical and electrochemical properties, a series of CNF-PDMS coated electrodes with different CNF/PDMS ratios (2, 4, 6 and 8 wt%) was fabricated. It was not possible to increase the doping level of paste with higher CNF content because of mixing issues (beyond a content higher than 8 wt%, the layer becomes heterogeneous). It is important to select the doping levels offering the lowest resistivity and the highest electrochemical reactivity.

#### 3.1.1. Electrical Characterization

Figure 2 shows the electrical conductivities of the CNF-PDMS composite layers as a function of doping levels. A significant decrease in resistivity can be clearly observed as the amount of CNF in the composite increases. Thus, the electrical conductivity of the CNF-PDMS composite films gradually increases approaching 80 S m^−1^ as the CNF concentration increases. Based on these results, the CNF-PDMS films show high electrical conductivity at 8 wt% of CNFs. This conductivity value is equivalent to the conductivity of carbon nanotubes/PDMS composites (100 S m^−1^:8 wt%) [21] and higher than that of the black carbon/PDMS composites (10 S m^−1^:25 wt%) previously described [22]. CNF-PDMS composite starts exhibiting electrical conductivity for low mass ratio of CNFs. These results indicate that the percolation is more efficient with the dimensions and the elongated shape of CNFs than with the spherical shape of the carbon black nanoparticles.

#### 3.1.2. Electrochemical Properties

Voltammetric study at different scan rates was performed in 0.1 M NaPBS pH 7.0 at CNF-PDMS composite as a function of the doping level, to determine the interfacial capacitance of electrodes. It results from the accumulation of charges on electrode surfaces during their polarization. This capacitance increases with the electroactive area of the geometrical surface. The interfacial capacitance was estimated from the ratio between measured current and the scan rate (see SI.2). An increase of the capacitance value from (10 ± 6) μF cm^−2^ to (159 ± 21) μF cm^−2^ was observed by increasing the doping level of CFN-PDMS composite from 2% to 8% (wt%) (Table 1). This reflects an increase of about 15 times of the electroactive surface. Assuming that the double layer capacitance is usually around 20 μF cm^−2^ in aqueous media [23], it was estimated that electroactive surface of 2, 4, 6 and 8% (wt%) of CNF-PDMS composite increased about 0.5, 1, 2 and 8 times, respectively, with respect to its geometrical area (Table 1). It is important to note that anode materials with high capacitance improve power output of MFCs, as previously reported [23,24,25]. 

Figure 3 shows the cyclic voltammograms obtained with CNF-PDMS composites at different doping levels after the addition of 1 mMof potassium ferrocyanide in a 0.1 M phosphate buffer solution of pH 7.4. The oxidation/reduction waves of ferricyanide/ferrocynide couple were observed in all voltammograms. It was observed that the peak potential difference decreased by increasing the CNF doping level. Besides, the oxidation and reduction peak currents increased with the CNF content. These results indicated that the ability of CNF-PDMS composite to catalyze the oxidation of ferrocianyde and the reduction of ferricyanide increased with the CNF content. The best electrocatalytic properties were obtained with the 8% (wt%) doped CNF-PDMS composite. This is due to a higher active surface compared to the same geometric surface.

#### 3.1.3. Morphological Analysis

The surface morphologies of CNF-PDMS at different doping levels were investigated by scanning electron microscope (SEM). As shown in Figure 4, the CNFs were homogeneously and randomly arranged at the composite surface. The SEM images clearly reveal that CNFs became more densely packed and more visible on the electrode surface by increasing CNF content. These images are similar to those observed by Al-Saleh et al. for carbon fibers trapped in polypropylene [26]. These results indicate that the high cross-linked structure of CNFs ensures the electrical contact between adjoining fibers despite the many void regions observed on the surface. Furthermore, the significant increase of electroactive surface with doping level can be explained by the comb-like structures of CNF on the surface. These comb-like conducting structures facilitate the direct extracellular electron transfer (EET) of electroactive bacteria (EAB) (e.g., *Geobacter sulfurreducens*) to the electrode, as previously reported [1]. Based on these results, it appears that the CNF-PDMS composite with 8% (wt%) doping level is the most suitable as an anode material for EAB growth in MFCs. Therefore, this doping level was chosen for coating Cu anodes in the rest of this study.

### 3.2. Anticorrosion Performance of CNF-PDMS Coated Copper Electrodes

To check the susceptibility of CNF-PDMS coated Cu electrodes to chemical corrosion, Cu foils modified with different thicknesses (250, 500 and 1000 µm) of 8 wt% CNF-PDMS composites, were prepared. The best corrosion resistance was obtained with 500 µm thick layer of CNF-PDMS. Indeed, Tafel curves indicated a significant decrease of corrosion current from (1.5 ± 0.5) mA.cm^−2^ to 8 ± 2 µA cm^−2^ after coating Cu electrodes with 500 µm of CNF-PDMS (Figure 5a). The anticorrosion efficiency of coating was supported by the increase of the polarization resistance (R_p_) of Cu electrodes from (3 ± 1) × 10^2^ Ω cm^−2^ to (50 ± 6) × 10^2^ Ω cm^−2^ after with 500 µm of CNF-PDMS. The corrosion rate of bare Cu electrodes decreased from (17 ± 6) × 10^3^ μm y^−1^ to (93 ± 23) μm y^−1^ after coating. Thus, the corrosion rate of bare Cu electrodes was about 180 times higher than that of the 500 µm-thick CNF-PDMS coated electrodes. These results indicate that CNF-PDMS layers prevent the direct contact between Cu surfaces and solutions. Figure 5b shows a small influence of coating thickness on anti-corrosion performance of CNF-PDMS layers. Corrosion currents of (30 ± 5) µA cm^−2^ and (10 ± 3) µA cm^−2^, and polarization resistances of (8 ± 1) × 10^2^ Ω cm^−2^ and (45 ± 15) × 10^2^ Ω cm^−2^, were obtained for Cu electrodes coated with 250 µm and 1000 µm thick layers, respectively. The corrosion rate of (351 ± 58) μm y^−1^, obtained with 250 µm thick layer was about three times higher than that obtained with 500 and 1000 µm ((117 ± 35) μm y^−1^) thick layers. These results indicate that optimal anti-corrosion performance was obtained from a CNF-PDMS layer thickness of 500 μm.

### 3.3. Electricity Production Performance

Voltage outputs of MFCs were recorded as a function of the time during two weeks in order to follow biofilm growth on unmodified Cu anodes and Cu anodes coated with different thicknesses (250, 500 and 1000 µm) of 8 wt% CNF-PDMS (Figure 6a). The voltage of the MFC based on unmodified Cu anodes increased quickly from the first day and then decreased to reach a negligible value after 3 days. This initial voltage increase is probably due to abiotic current generation from Cu galvanic corrosion, as previously described [16]. Thereafter, the voltage decrease suggests that chemical corrosion of Cu anodes leads to production of copper oxides, which formed passivation layer on Cu surface [27]. MFCs based on Cu-modified anodes showed an acclimation time of around a week before starting current production. Voltage outputs started to increase at the 5th day of the experiment and became stable after 10 days. These results indicate that growth kinetics of biofilms on coated anodes are almost the same for different CNF-PDMS layer thickness. The startup time of MFC with CNF-PDMS anodes was lower than that reported for MFCs with conventional carbon-based anodes [27,28,29]. This lag phase in which no current had been observed was probably increased by the hydrophobic character of PDMS present on the surface. Indeed, hydrophobic surfaces slowed down adhesion of bacteria and made the startup phase of MFC longer, as previously reported [30].

Figure 6b shows polarization curves of MFCs obtained with unmodified and modified Cu anodes after 2 weeks of operation. The performances of MFCs are presented in Appendix A. The maximum power density of (8 ± 1) mW/m² obtained with bare Cu anodes was significantly lower than that attained in the MFCs using coated Cu anodes. The best performance was obtained with 500 µm-thick CNF-PDMS coated electrodes with a maximum power density of (70 ± 8) mW m^−2^, which was about eight times higher than that of the uncoated electrodes. The maximum power density obtained with 250 µm and 1000 µm thicknesses were (35 ± 4) mW m^−2^ and (50 ± 3) mW m^−2^, respectively. These power densities were lower than that obtained with 500 µm-thick CNF-PDMS coated electrodes. These results suggest that electroactive biofilms formed on the thinner and thicker layers of CNF-PDMS were less efficient than that obtained on 500 µm-thick layers. The difference in performance of CNF-PDMS layers depending on thickness could be related to their anti-corrosion performances described above. Indeed, corrosion rate determine the amount of soluble Cu produced at the surface of coated electrodes, which inhibited more or less bacterial activity and consequently reduced current generation by electroactive bacteria. The power density of 500 µm-thick CNF-PDMS coated electrodes was normalized to the volume of the anodes, in order to compare their performances with those of polyaniline-Cu anodes previously described by Mwale et al., [17]. The normalized power density of CNF-PDMS coated anodes was around 70 W m^−3^, which was about 10^4^ times higher than that of polyaniline-Cu anodes ((1.5 ± 0.3) mW m^−2^). Since authors used the same inoculums (wastewater) to operate MFCs, this important difference in electrical performances can be explained either by the low conductivity of polyaniline layers or a less efficient electron transfer between electroactive bacteria and functionalized Cu electrodes, compared to CNF-PDMS coated electrodes. Indeed, the authors justified this poor electrical performance by the low conductivity of the polyaniline and the compact passivated layer (oxide layer) formed between the polymer-Cu surfaces. Compared with the performance of MFCs with other anode materials using wastewater as inoculum, the power density of 500 µm-thick CNF-PDMS coated Cu anodes was higher than what we previously reported for stainless steel anodes coated with CNF-PDMS (19 mW m^−2^) [19] and bare carbon clothe (50 mW m^−2^) [1]. However, the power density was lower than those obtained with carbon felt (100 mW m^−2^) [29] and carbon paper (104 mW m^−2^) [28] anodes, as previously described. Higher power densities (1303 mW m^−2^) were reported for carbon felt anodes functionalized with electroactive dopants [29]. However, this surface functionalization method suffers from limited processibility because it cannot be deposited on large electrode surfaces at low costs.

To assess electron transfer reactions on uncoated and coated Cu electrodes, cyclic voltammograms of MFCanodes were recorded after 2 weeks of operation (Figure 7). Figure 7a shows that bare Cu anode has a large cathodic peak at −270 mV (vs AgCl/Ag). It can be attributed to reduction of copper oxide, as previously described [31]. In the positive sweep, high oxidation current at 250 mV (vs. AgCl/Ag) was recorded, which is attributed to the oxidation of metallic copper. Cyclic voltammograms of coated anodes showed markedly reduced anodic and cathodic currents, indicating decreased electrochemical activity of interface after coating. These results confirm anti-corrosion performance of CNF-PDMS layers. Cyclic voltammograms of coated Cu anodes exhibited anoxidation/reduction peaks centered at −300 mV (vs. Ag/AgCl). Based on previously reported values, the negative potential region of these redox peaks corresponded to electron transfer of electroactive bacteria, such as *Shewanella oneidensis* and *Geobacter sulfurreducens* [32,33]. Cyclic voltammograms of CNF-PDMS-Cu anodes showed a higher electroactivity with higher faradaic peaks for biofilms formed on 500 µm-thick CNF-PDMS surfaces. These results are in agreement with the power performance of MFCs. Based on all these results, 500 µm-thick CNF-PDMS coating provides the best performances in term of corrosion protection of Cu electrodes and in term of electroactive biofilm growth, as well as generation of electrical power in MFC application.

### 3.4. Stability of CNF-PDMS Coated Copper Electrodes

Figure 8 shows microscope images of uncoated and 500 µm-thick coated Cu anodes of MFCs after 2 weeks of operation. Figure 8a showed that Cu coated anode remained intact, after prolonged exposure of the electrode in wastewater solution, without undergoing surface degradation. Indeed, no significant change or noticeable damage was observed on Cu surface after removing of CNF-PDMS coating (Figure 8b). While microscope images of bare Cu anode showed important degradation of surfaces with fracture formation (Figure 8c). A thin black layer can be clearly observed on the surface of Cu foil, which can be due to the formation of passivated oxide layer, as previously described [34].

### 3.5. Economic and Technologie Considerations

The cost of anode materials is important for the practical application of MFCs. Table 2 compares the prices of some examples of flat conducting materials commonly used as anodes in MFCs. The major point of interest of this analysis, is only based on the material price and does not take into account the processing costs that have to be included for a more detailed analysis. The cost of Cu foil anodes is cheaper than most of these flat carbon anodes. In addition, the functionalization of a plane electrode surface with 500 µm of CNF-PDMS layer costs USD 150 per m^2^. The cost per watt (CPW) of anode materials was calculated by dividing the cost of material by power density obtained with corresponding MFCs. The CPW is a simple measurement that can be used to compare the price/performance ratio of anode materials in the prospect of estimating the cost of capital. Materials with a lower CPW are more interesting for scaling-up applications of MFCs. The CPW of CNF-PDMS-Cu anodes (EUR 3.2 k W^−1^) is around 23, 5 and 3 times lower than that of CNF-PDMS-SS anodes, carbon paper and carbon cloth, respectively (Table 2). It is almost similar to the lower CPW of carbon felt anodes. However, CNF-PDMS-Cu anodes are more interesting than carbon felt for the scale-up of MFCs. Indeed, carbon felt needs metallic collectors for electrical connection to the external circuit that could be a source of potential loss for large carbonaceous electrode surfaces due to the low electrical conductivity of carbon material and the long distance traveled by electrons to the collector [15]. In the case of CNF-PDMS-Cu anodes, Cu plays the role of the collector and the distance traveled by the electrons (500 µm thick layer) is the same whatever the dimensions of the electrodes.

## 4. Conclusions

The present work described a novel CNF-PDMS coated Cu electrodes that can be used as anodes in MFC application and can be employed for different electrochemical applications in aqueous electrolytes. These electrodes were fabricated using a simple process that can be produced by the roll-to-roll method on flexible Cu supports. It was found that CNF-PDMS composites with a higher CNF percentage (8 wt%) showed the highest conductivity and best electrochemical activity. Indeed, the comb-like structure of CNF-PDMS surfaces increased electroactive surface of coated electrodes. Moreover, this coating layer drastically decreased the corrosion rate of Cu electrodes. The optimal anti-corrosion performance was obtained with a CNF-PDMS layer thickness of 500 μm. This coating thickness also allowed the growth of electroactive biofilms providing (70 ± 8) mW m^−2^ of power density in MFC application. This electrical performance is 10^4^ times higher than that of polyaniline coated Cu anodes reported in the only study on surface modification using Cu material as an anode in MFC operations. Chemical stability, biocompatibility and low-cost of CNF-PDMS-coated Cu electrodes suggest that this material is a prospective candidate as an anode in MFC application.

## Figures and Tables

**Figure 1 nanomaterials-11-03144-f001:**
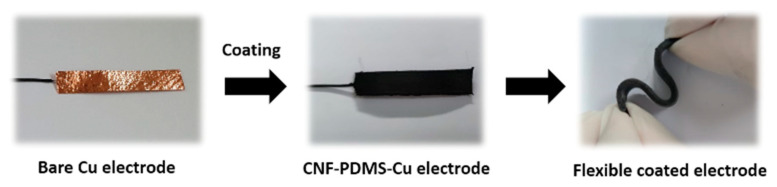
Images of uncoated and coated Cu electrodes.

**Figure 2 nanomaterials-11-03144-f002:**
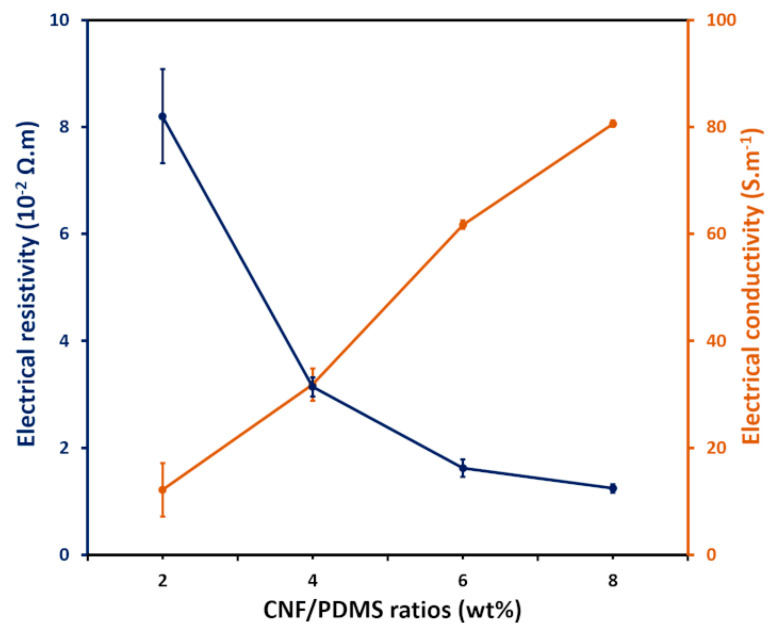
Resistivity (blue) and conductivity (orange) of CNF-PDMS composite versus CNF weight concentration.

**Figure 3 nanomaterials-11-03144-f003:**
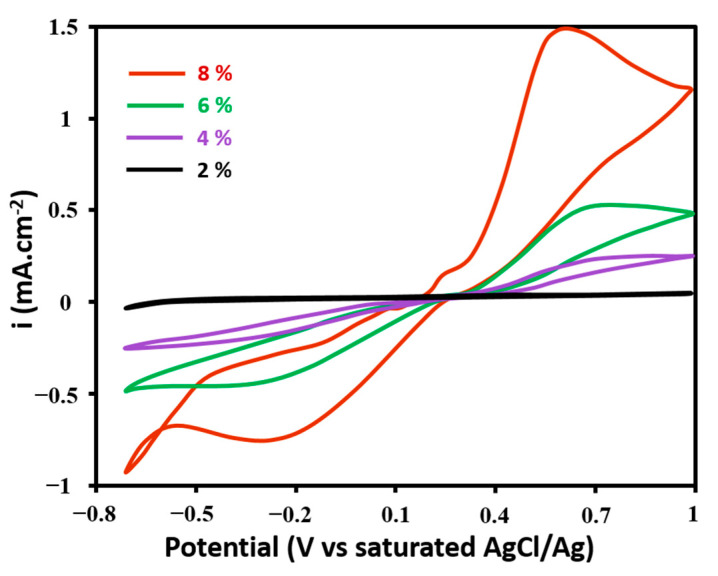
CVs of CNF-PDMS composites at different doping levels in presence of 1 mM of potassium ferrocyanide in 0.1 M phosphate buffer solution.

**Figure 4 nanomaterials-11-03144-f004:**
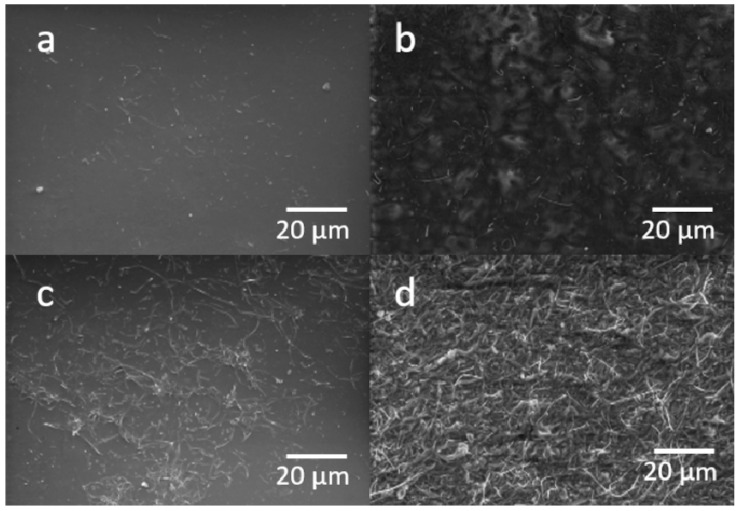
SEM images of CNF-PDMS composites at 2% (**a**), 4% (**b**), 6% (**c**) and 8% (**d**) doping levels.

**Figure 5 nanomaterials-11-03144-f005:**
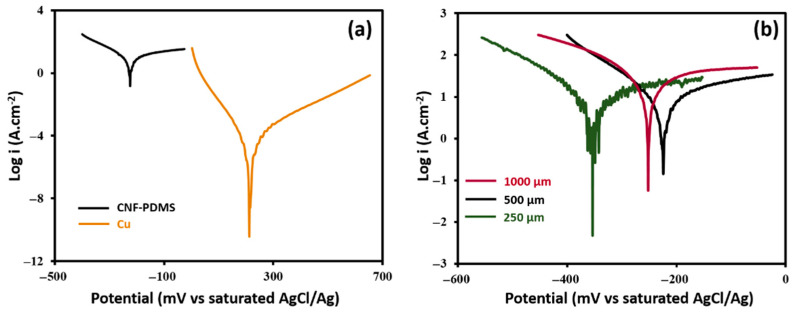
(**a**) Tafel curves of uncoated Cu electrode (orange curve) and 500 µm-thick CNF-PDMS-Cu coated electrodes (black curve). (**b**) Tafel curves of Cu coated electrodes with 250 µm (green curve), 500 µm (black curve) and 1000 µm (granite curve) thicknesses of 8 wt% CNF-PDMS layers.

**Figure 6 nanomaterials-11-03144-f006:**
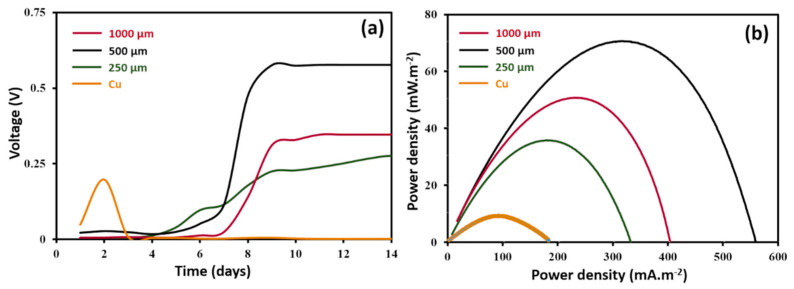
Voltage as a function of time (**a**) and polarization curves (**b**) of MFC based on unmodified Cu anode (orange curve) and with Cu anodes coated with 250 µm (green curve), 500 µm (black curve) and 1000 µm (granite curve) thicknesses of 8 wt% CNF-PDMS layers.

**Figure 7 nanomaterials-11-03144-f007:**
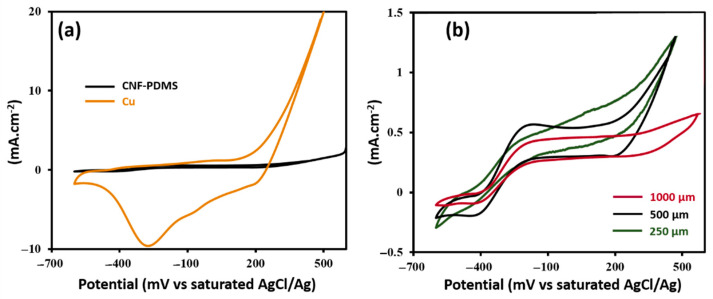
(**a**) CVs of uncoated Cu electrode (orange curve) and 500 µm-thick CNF-PDMS-Cu coated electrodes (black curve). (**b**) CVs of Cu coated electrodes with 250 µm (green curve), 500 µm (black curve) and 1000 µm (granite curve) thicknesses of 8 wt% CNF-PDMS layers.

**Figure 8 nanomaterials-11-03144-f008:**
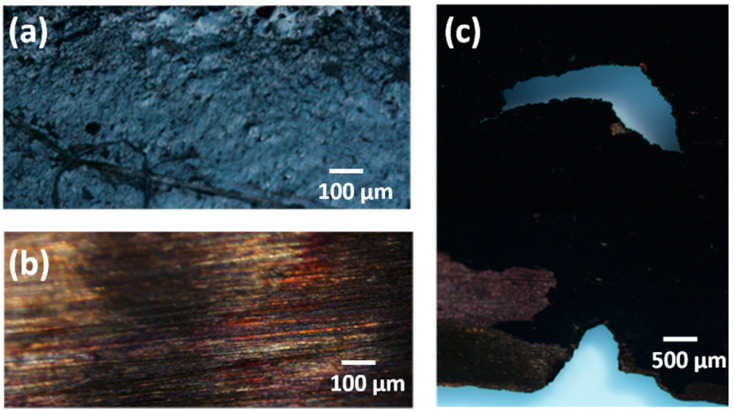
Light microscope images of (**a**): 500 µm-thick CNF-PDMS coated Cu anode; (**b**): Cu anode after removing coating; (**c**): Bare Cu anode.

**Table 1 nanomaterials-11-03144-t001:** Values of interfacial capacitance and electroactive/geometrical area ratio for different CNF/PDMS ratios.

CNF/PDMS Ratio (wt%)	Interfacial Capacitance (µF cm^−2^)	Electroactive Surface/Geometrical Area (%)
2	10 ± 6	50 ± 30
4	25 ± 17	125 ± 85
6	52 ± 20	260 ± 100
8	159 ± 21	795 ± 105

**Table 2 nanomaterials-11-03144-t002:** Characteristics of flat materials used as anodes in MFCs.

Anode Materials	Maximum Power Density (mW m^−2^)	References	Unite Price (EUR.m^−2^)	Price Per Watt (EUR k W^−1^)
Carbon felt	100	[30]	300 ^a^	3
Carbon paper	104	[29]	1700 ^b^	16.3
Carbon clothe	50	[28]	500 ^b^	10
Stainless Steel (SS) plate	4	[19]	300 ^c^	75
CNF-PDMS-SS plate	19	[19]	550 ^d,e^	20
Cu foil	8	This work	80 ^c^	10
CNF-PDMS-Cu foil	70	This work	230 ^c,d,e^	3.2

^a^ 2021 values from https://www.graphitech-usinage.fr; ^b^ 2021 values from https://www.goodfellow.com; ^c^ 2021 values from https://fr.rs-online.com/; ^d^ 2021 values from https://www.neyco.fr/; ^e^ 2021 values from https://www.sigmaaldrich.com.

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
