# Peer review of "Carbon Nano-Fiber/PDMS Composite Used as Corrosion-Resistant Coating for Copper Anodes in Microbial Fuel Cells"

_nanomaterials, 2021, doi:10.3390/nano11113144_

Round 1
Reviewer 1 Report
The paper discusses the research work on Cu anode for the development of high performance anode materials of microbial fuel cells. The novelty of the paper, as described by the authors, lies in the surface modification of Cu anode to change its raw state to be more suitable for MFC application.Based on the developed coating on Cu anode, the anode can get the highest conductivity and the best electrochemical activity. The results of this work suggests that it will be a promising anode in MFC application. - Introduction is very well-written. Flow was good and research gaps were appropriately discussed and presented- The study lacks appropriate discussion about how some result of this study is compared to other studies in literature. This comment persists in the electricity production part 3.3. - The electron transfer resistance of anode with and without the coating should be given, it is very important to assess the electron transfer reactions on Cu anodes.h the literature. However, the table is selective and not consistent. Not all studies discussed in the introduction were included in the table. Also, 50% of the mentioned studies were done using mixed culture which makes the com
Author Response
- The study lacks appropriate discussion about how some result of this study is compared to other studies in literature. This comment persists in the electricity production part 3.3.
Answer:
We agree with Reviewer #1. We compared the result of this study with those of flat conducting materials commonly used in MFCs. This comparison concerned the biofilm growth and the electricity production in the part 3.3. This comparison is added as following in a paragraph 3.3:
Page 8, starting line 270:
The startup time of MFC with CNF-PDMS anodes was lower than that reported for MFCs with conventional carbon-based anodes [27]–[29]. This lag phase in which no current had been observed, was probably increased by the hydrophobic character of PDMS present on the surface. Indeed, hydrophobic surfaces slowed down adhesion of bacteria and made the startup phase of MFC longer, as previously reported [30].
Page 9, starting line 309:
Compared with the performance of MFCs with other anode materials using wastewater as inoculum, the power density of 500 µm-thick CNF-PDMS coated Cu anodes was higher than that we previously reported for stainless steel anodes coated with CNF-PDMS (19 mW.m-2) [18] and bare carbon clothe (50 mW.m-2) [31]. However, the power density was lower than those obtained with carbon felt (100 mW.m-2) [29] and carbon paper (104 mW.m-2) [28] anodes, as previously described. Higher power densities (1303 mW.m−2) were reported for carbon felt anodes functionalized with electroactive dopants [29]. However, this surface functionalization method suffers from limited processibility because it can not be deposited on large electrode surfaces at low costs.
- The electron transfer resistance of anode with and without the coating should be given, it is very important to assess the electron transfer reactions on Cu anodes.h the literature.
We added the polarization resistance of Cu anodes with and without coating as following in a paragraph 3.2:
Page 7, starting line 238:
The anticorrosion efficiency of coating was supported by the increase of the polarization resistance (Rp) of Cu electrodes from (3 ± 1) x102 Ω.cm−2 to (50 ± 6) x102 Ω.cm−2 after with 500 µm of CNF-PDMS.
Page 7, starting line 246:
Corrosion currents of (30 ± 5) µA.cm-2 and (10 ± 3) µA.cm-2, and polarization resistances of (8 ± 1) x102 Ω.cm−2 and (45 ± 15) x102 Ω.cm−2, were obtained for Cu electrodes coated with 250 µm and 1000 µm thick layers, respectively.
- However, the table is selective and not consistent. Not all studies discussed in the introduction were included in the table. Also, 50% of the mentioned studies were done using mixed culture which makes the com
Answer: We modified the table 2 and the paragraph of the part 3.5 as following:
Page 11, starting line 358:
The major point of interest of this analysis, is only based on the material price and does not take into account the processing costs that have to be included for a more detailed analysis. The cost of Cu foil anodes is cheaper than most these flat carbon anodes. In addition, the functionalization of a plane electrode surface with 500 µm of CNF-PDMS layer costs 150 $ per m2. The cost per watt (CPW) of anode materials was calculated by dividing the cost of material by power density obtained with corresponding MFCs. The CPW is a simple measurement that can be used to compare the price/performance ratio of anode materials in the prospect of estimating the cost of capital. Materials with a lower CPW are more interesting for scaling-up applications of MFCs. The CPW of CNF-PDMS-Cu anodes (3.2 k€.W-1) is around 23, 5 and 3 times lower than that of CNF-PDMS-SS anodes, carbon paper and carbon cloth, respectively (Table. 2). It is almost similar to the lower CPW of carbon felt anodes. However, CNF-PDMS-Cu anodes are more interesting than carbon felt for the scale-up of MFCs. Indeed, carbon felt needs metallic collectors for electrical connection to the external circuit that could be a source of potential loss for large carbonaceous electrode surfaces due to the low electrical conductivity of carbon material and the long distance traveled by electrons to the collector [14]. In the case of CNF-PDMS-Cu anodes, Cu plays the role of the collector and the distance traveled by the electrons (500 µm thick layer) is the same whatever the dimensions of the electrodes.
Table 2. Characteristics of flat materials used as anodes in MFCs.
|
Anode materials |
Maximum power density (mW.m-2) |
Reference |
Unite price (€.m-2) |
Price per Watt (k€.W-1) |
|
Carbon felt |
100 |
[29] |
300a |
3 |
|
Carbon paper |
104 |
[28] |
1700b |
16.345 |
|
Carbon clothe |
50 |
[27] |
500b |
10 |
|
Stainless Steel (SS) plate |
4 |
[18] |
300c |
75 |
|
CNF-PDMS-SS plate |
19 |
[18] |
550 d,e |
20 |
|
Cu foil |
8 |
This work |
80c |
10 |
|
CNF-PDMS-Cu foil |
70 |
This work |
230c,d,e |
3.2 |
a: 2021 values from http://www.graphitech-usinage.fr
b: 2021 values from https://www.goodfellow.com
c: 2021 values from https://fr.rs-online.com/
d: 2021 values from https://www.neyco.fr/
e: 2021 values from https://www.sigmaaldrich.com

Reviewer 2 Report
The design, composition and preparation of anodes are important factors in microbial fuel cells. It is important that carbon nanofibers coated by polydimethylsiloxane is employed as a corrosion-resistant anode. However, the coating may affect the catalytic performance of anodes. This work reported a more comprehensive and detailed study on the formation and electrochemical performance of polydimethylsiloxane coating. The conclusions obtained have certain reference significance for practical microbial fuel cells. Therefore, I would like to recommend the publication of this work after the following issues are made:
- In the introduction, some pertinent publications on electrode design (ChemElectroChem, 2021, 8, 2583-2589) and reasonable design of structure and composition playing a key role in achieving high performance (Bioelectrochemistry, 2021, 137, 107675) are to be cited.
- What is the semi-quantitative relationship between catalysis and corrosion resistance based on different polydimethylsiloxane film thickness?
- What effect does the film have on device startup time and bacterial compatibility?
- Pay attention to the standardization of language expression.
Author Response
Reviewer #2:
- In the introduction, some pertinent publications on electrode design (ChemElectroChem, 2021, 8, 2583-2589) and reasonable design of structure and composition playing a key role in achieving high performance (Bioelectrochemistry, 2021, 137, 107675) are to be cited.
Answer: We added the reference Bioelectrochemistry, 2021, 137, 107675 in the introduction (Reference 7).
- What is the semi-quantitative relationship between catalysis and corrosion resistance based on different polydimethylsiloxane film thickness?
Answer:
We added the polarization resistance of Cu anodes without and with coating for different film thickness as following in a paragraph 3.2:
Page 7, starting line 238:
The anticorrosion efficiency of coating was supported by the increase of the polarization resistance (Rp) of Cu electrodes from (3 ± 1) x102 Ω.cm−2 to (50 ± 6) x102 Ω.cm−2 after with 500 µm of CNF-PDMS.
Page 7, starting line 246:
Corrosion currents of (30 ± 5) µA.cm-2 and (10 ± 3) µA.cm-2, and polarization resistances of (8 ± 1) x102 Ω.cm−2 and (45 ± 15) x102 Ω.cm−2, were obtained for Cu electrodes coated with 250 µm and 1000 µm thick layers, respectively.
We were enable to determine a semi-quantitative relationship between catalysis and corrosion resistance based on different polydimethylsiloxane film thickness.
- What effect does the film have on device startup time and bacterial compatibility?
Answer:
We compared the biofilm growth in this study with those of flat conducting materials commonly used in MFCs. This comparison is added as following in a paragraph 3.3:
Page 8, starting line 270:
The startup time of MFC with CNF-PDMS anodes was lower than that reported for MFCs with conventional carbon-based anodes [27]–[29]. This lag phase in which no current had been observed, was probably increased by the hydrophobic character of PDMS present on the surface. Indeed, hydrophobic surfaces slowed down adhesion of bacteria and made the startup phase of MFC longer, as previously reported [30].
- Pay attention to the standardization of language expression.
Answer:
A thorough re-reading of this article was done

Round 2
Reviewer 2 Report
It can be published.